# ADAPTIVE GRADIENT METHODS CAN BE PROVABLY FASTER THAN SGD WITH RANDOM SHUFFLING

## ABSTRACT

Adaptive gradient methods have been shown to outperform SGD in many tasks of training neural networks. However, the acceleration effect is yet to be explained in the non-convex setting since the best convergence rate of adaptive gradient methods is worse than that of SGD in literature. In this paper, we prove that adaptive gradient methods exhibit an $\tilde{O}(T^{-1/2})$-convergence rate for finding first-order stationary points under the strong growth condition, which improves previous best convergence results of adaptive gradient methods and random shuffling SGD by factors of $O(T^{-1/4})$ and $O(T^{-1/6})$, respectively. In particular, we study two variants of AdaGrad with random shuffling for finite sum minimization. Our analysis suggests that the combination of random shuffling and adaptive learning rates gives rise to better convergence.

## 1    INTRODUCTION

We consider the finite sum minimization problem in stochastic optimization:

$$\min_{\boldsymbol{x}\in\mathbb{R}^d} \quad f(\boldsymbol{x}) = \frac{1}{n}\sum_{i=1}^{n} f_i(\boldsymbol{x}), \tag{1}$$

where $f$ is the objective function and its component functions $f_i : \mathbb{R}^d \rightarrow \mathbb{R}$ are smooth and possibly non-convex. This formulation has been used extensively in training neural networks today. Stochastic gradient descend (SGD) and its variants have shown to be quite effective for solving this problem, whereas recent works demonstrate another prominent line of gradient-based algorithms by introducing adaptive step sizes to automatically adjust the learning rate (Duchi et al., 2011; Tieleman & Hinton, 2012; Kingma & Ba, 2014).

Despite the superior performance of adaptive gradient methods in many tasks (Devlin et al., 2019; Vaswani et al., 2017), their theoretical convergence remains the same or even worse for non-convex objectives, compared to SGD. In general non-convex settings, it is often impractical to discuss optimal solutions. Therefore, the attention of analysis turns to stationary points instead. Many works have been proposed to study first-order (Chen et al., 2019; Zhou et al., 2018; Zaheer et al., 2018; Ward et al., 2018; Zhou et al., 2018) and second-order (Allen-Zhu, 2018; Staib et al., 2019) stationary points. Table 1 summarized some previous best-known results for finding first-order stationary points. One might notice that the best dependence on the total iteration number $T$ of adaptive gradient methods matches that of vanilla SGD. In addition, with the introduction of incremental sampling techniques, an even better convergence of SGD can be obtained (Haochen & Sra, 2019; Nguyen et al., 2020).

This gap between theory and practice of adaptive gradient methods has been an open problem that we aim to solve in this paper. Motivated by the analysis of sampling techniques, we rigorously prove that adaptive gradient methods exhibit a faster non-asymptotic convergence rate that matches the best result on SGD. In particular, we make the following contributions:

- Our main contribution (Theorem 1,2,3) is to prove that two variants of AdaGrad can find $\tilde{O}(T^{-1/2})$-approximate first-order stationary points under the strong growth condition assumption (Schmidt & Roux, 2013). This improves previous best convergence results of adaptive gradient methods and shuffling SGD by factors of $O(T^{-1/4})$ and $O(T^{-1/6})$ ,

Table 1: Convergence rate comparisons for the non-convex optimization problem. First two categories, 'SGD' and 'Adaptive Gradient Methods' are based on the *expectation minimization* problem, whereas the last category is based on the *finite sum minimization* problem.

| | Algorithm | Assumptions (L-smooth+) | $\|\nabla f(\boldsymbol{x})\|$-convergence $T$ |
|---|---|---|---|
| **SGD** | vanilla SGD | • $\sigma^2$ bounded gradient variance | $O\left(T^{-1/4}\right)$ |
| | Constant step-size SGD (Vaswani et al., 2019) | • strong growth condition | $O\left(T^{-1/2}\right)$ |
| **Adaptive Gradient Methods** | AMSGrad, AdaFom (Chen et al., 2019) | • bounded gradients 
 • initial gradient lower bound | $\tilde{O}\left(T^{-1/4}\right)$ |
| | AMSGrad, Padam (Zhou et al., 2018) | • bounded gradients 
 • gradient sparsity | $O\left(T^{-1/4}\right)$ |
| | RMSProp, Yogi (Zaheer et al., 2018) | • bounded gradients 
 • $\sigma^2$ bounded gradient variance | $O\left(T^{-1/2} + \sigma\right)$ |
| | AdaGrad-NORM (Ward et al., 2018) | • bounded gradients 
 • $\sigma^2$ bounded gradient variance | $\tilde{O}\left(T^{-1/4}\right)$ |
| | GGT (Agarwal et al., 2019) | • $\sigma^2$ bounded gradient variance | $O\left(T^{-1/4}\right)$ |
| **Shuffling** | Random Shuffling SGD (Nguyen et al., 2020) | • bounded gradients | $\tilde{O}\left(n^{1/3} \cdot T^{-1/3}\right)$ |
| | Ours | • bounded gradients 
 • strong growth condition | $\tilde{O}\left(n^{1/2} \cdot T^{-1/2}\right)$ |

$T$ denotes the number of iterations; $n$ is the number of samples in the finite sum minimization problem.

respectively. As a result, this bridges the gap between analysis and practice of adaptive gradient methods by proving that adaptive gradient methods can be faster than SGD with random shuffling in theory.

- We study the strong growth condition under which our convergence rate is derived. This condition has been previous used to study SGD in the expectation minimization setting (Schmidt & Roux, 2013; Vaswani et al., 2019). We prove that this condition is satisfied by two general types of models under some additional assumptions.

- We conduct preliminary experiments to demonstrate the combined acceleration effect of random shuffling and adaptive learning rates.

Our analysis points out two key components that lead to better convergence results of adaptive gradient methods: the epoch-wise analysis of random shuffling can incorporate the benefit of full gradients; the adaptive learning rates along with the strong growth condition provide better improvement of objective value in consecutive epochs.

**Finite Sum Minimization vs. Expectation Minimization** The comparison in Table 1 shows the convergence rates in the non-convex setting with respect to first-order stationary points. The results in the first two categories apply to the general expectation minimization problem with $f(\boldsymbol{x}) = \mathbb{E}_{\boldsymbol{z}} f(\boldsymbol{x}, \boldsymbol{z})$. Whereas the convergences for expectation minimization naturally transform to finite sum minimization, the statements remain asymptotic, meaning $\mathbb{E}\|\nabla f(\boldsymbol{x})\| \sim O(T^\alpha)$, where the expectation is taken to compensate the stochastic gradients. Many efforts have been made to reduce variance in finite sum minimization (Johnson & Zhang, 2013; Reddi et al., 2016; Haochen & Sra, 2019). In particular, non-asymptotic results can be gained using random shuffling, under which the dependency on sample size $n$ seems to be unavoidable (Haochen & Sra, 2019).

## 2 PRELIMINARIES

A typical setting of machine learning using gradient methods is the finite sum minimization in equation (1). In this problem, the number of samples $n$ is usually very large, rendering the evaluation of full gradients expensive. Therefore, a mini-batch gradient is introduced to approximate the full

gradient. Mini-batch gradient descent is often carried out in epochs, where each epoch includes several iterations of parameter updates. This epoch-wise implementation can easily incorporate shuffling techniques, which have proven to be effective for SGD both in theory and practice.

We aim to analyze the convergence rate of adaptive gradient methods under this framework, where the objective can be non-convex. Throughout this paper, we restrict the discussions of convergence to achieving $\epsilon$-approximate first-order stationary point defined as $\boldsymbol{x}$ satisfying $\|\nabla f(\boldsymbol{x})\| \leq \epsilon$. We leave for future work analysis related to saddle points and second-order stationary points. We want to show that adaptive gradient methods can find $\boldsymbol{x}$ such that $\|\nabla f(\boldsymbol{x})\| = \tilde{O}(T^{-1/2})$ in $T$ epochs.

**Notations.** $\boldsymbol{v}^2$ denotes the matrix $\boldsymbol{v}\boldsymbol{v}^\top$ and $\|\boldsymbol{v}\|$ is the $l_2$-norm of vector $\boldsymbol{v}$; $\mathrm{diag}(\boldsymbol{V})$, $\|\boldsymbol{V}\|$, $\lambda_{\min}(\boldsymbol{V})$ and $\lambda_{\max}(\boldsymbol{V})$ are the diagonal matrix, the spectral norm, the largest and smallest non-zero eigenvalues of the matirx $\boldsymbol{V}$, respectively. For alphabets with subscripts, $\boldsymbol{v}_{i:j}$ denotes the collection of $\{\boldsymbol{v}_i, \boldsymbol{v}_{i+1}, ..., \boldsymbol{v}_j\}$ and $\boldsymbol{v}_:$ denotes the entire set of $\boldsymbol{v}_.$; similar notations are used for alphabets with double subscripts. Let $[n] = \{1, ..., n\}$, $O(\cdot), \tilde{O}(\cdot)$ be the standard asymptotic notations. Denote $\boldsymbol{e}_i$ as the unit vector with its $i$-th component being $1$ and $\boldsymbol{e}$ the all-one vector whose dimension depend on the context. As a clarification, we use $T$ to denote the number of epochs (instead of the number of iterations in Table 1) starting from this section.

**AdaGrad-type methods.** As opposed to SGD, adaptive gradient methods assign a coordinate-wise adaptive learning rate to the stochastic gradient. We formulate the generic AdaGrad-type optimizers, including their full and diagonal versions, as follows. At the $i$-th iteration of epoch $t$, the parameter is updated by:

$$\boldsymbol{x}_{t,i+1} = \boldsymbol{x}_{t,i} - \eta_t \boldsymbol{V}_{t,i}^{-1/2}\boldsymbol{g}_{t,i}, \qquad\qquad \text{(full version)}$$

$$\boldsymbol{x}_{t,i+1} = \boldsymbol{x}_{t,i} - \eta_t \mathrm{diag}(\boldsymbol{V}_{t,i})^{-1/2}\boldsymbol{g}_{t,i}, \qquad\qquad \text{(diagonal version)}$$

where $\boldsymbol{g}_{t,i}$ is the mini-batch gradient of the objective at $\boldsymbol{x}_{t,i}$, the matrix $\boldsymbol{V}_{t,i}$ contains second-moment calculated using all the past stochastic gradients and $\eta_t$ is the step size of epoch $t$. The initial parameter of epoch $t + 1$ is taken to be the parameter updated by epoch $t$, i.e. $\boldsymbol{x}_{t+1,1} = \boldsymbol{x}_{t,m+1}$, where we have $m$ iterations in each epoch. The full version is impractical for high-dimensional $\boldsymbol{x}$. Thus the diagonal version is often preferred in literature. As an example, the second-moment matrix in AdaGrad is taken to be $\boldsymbol{V}_{t,i} = (\sum_{s=1}^{t-1}\sum_{j=1}^{m}\boldsymbol{g}_{s,j}^2 + \sum_{j=1}^{i}\boldsymbol{g}_{t,j}^2)/t$. SGD can also be written into this general form where we set $\boldsymbol{V}_{t,i}$ to be the identity matrix.

**Sampling Strategy.** Random shuffling, also known as sampling without replacement, is an often-used technique to accelerate the convergence of SGD. The idea is to sample a random permutation of function indices $[n]$ for each epoch and slide through this permutation to get the mini-batch gradients for the iterations in this epoch. Some implementations shuffle the set $[n]$ uniformly independently for each epoch while others shuffle the set once during initialization and use the same permutation for all epochs. Generally speaking, suppose we have a permutation $\sigma = (\sigma_1, ..., \sigma_n)$ at epoch $t$, we define the set $\mathbb{B}_{t,i} = \{\sigma_j : (i-1)\frac{n}{m} < j \leq i\frac{n}{m}\}$ where $m$ is the number of iterations in one epoch. Then the mini-batch gradient is taken to be $\boldsymbol{g}_{t,i} = m/n \cdot \sum_{j \in \mathbb{B}_{t,i}} \nabla f_j(\boldsymbol{x}_{t,i})$.

This sampling method of mini-batches benefits the theoretical analysis of SGD by providing a bounded error between the full gradient and the aggregation of mini-batch gradients in one epoch (Haochen & Sra, 2019; Nguyen et al., 2020). A naive observation that backups this point can be made by assuming $\boldsymbol{x}_{t,1} = ... = \boldsymbol{x}_{t,m}$, since $\cup_{i=1}^{m}\mathbb{B}_{t,i} = [n]$, we would have $\sum_{i=1}^{m} \boldsymbol{g}_{t,i} = \nabla f(\boldsymbol{x}_{t,1})$. Then full gradient can be used to obtain convergence better than plain SGD.

**Random shuffling for AdaGrad.** Unlike SGD, in adaptive methods, it is hard to approximate the full gradient with the aggregation of mini-batch gradient updates in one epoch due to the presence of the second moments. As we will show in experiments, the simple shuffling variant that only changes the sampling method of mini batches in AdaGrad does not lead to better convergence. The major difficulty hampering the analysis of this variant is that the second-moment matrix uses all the gradient information in history without distinguishment. Thus to be able to leverage the benefit of the full gradient, we propose to study a slight modification of AdaGrad. Formally, we shuffle the set $[n]$ once at initialization and obtain the mini-batch gradients in a random shuffling manner. We update the

parameters by the same rules of AdaGrad-type methods described above where the second-moment matrix is taken to be:

$$\boldsymbol{V}_{t,i} = \sum_{j=i+1}^{m} \boldsymbol{g}_{t-1,j}^2 + \sum_{j=1}^{i} \boldsymbol{g}_{t,j}^2. \qquad \text{(AdaGrad-window)}$$

The difference between AdaGrad-window and AdaGrad is that the former only use the latest $m$ mini-batch gradients instead of an epoch-wise average of all the mini-batch gradients in history. The step size is $\eta_t = \eta/\sqrt{t}$ where $\eta$ is a constant for both methods. The updates of AdaGrad-window is also very similar to the GGT method (Agarwal et al., 2019) without momentum. However, GGT uses the full matrix inversion, where our analysis applies to both full and diagonal versions.

## 3 MAIN RESULTS

We will show that AdaGrad-window has the convergence rate of $\tilde{O}(T^{-1/2})$ for non-convex problems under some mild assumptions. This is a significant improvement compared with previous best convergence results of adaptive gradient methods and random shuffling SGD, which are of order $O(T^{-1/4})$ and $\tilde{O}(T^{-1/3})$ respectively. The *key* towards our convergence rate improvement is two-fold: the epoch-wise analysis of random shuffling enables us to leverage the benefit of full gradients; the adaptive learning rates and the strong growth condition endow a better improvement of objective value in consecutive epochs.

In order to achieve this better convergence, we first state the assumptions and important concepts used in the proof. Apart from the general assumptions (A1) and (A2) used in previous analysis (Fang et al., 2018; Zaheer et al., 2018; Ward et al., 2018), we pose another assumption described below in (A3) to characterize the consistency between individual gradients and the full gradient.

**Assumptions.** We assume the following for AdaGrad-window:

(A1) The objective function is lower bounded and component-wise $L$-smooth, i.e. $\exists f^* \in \mathbb{R}$ s.t. $f(\boldsymbol{x}) \geq f^* > -\infty, \forall \boldsymbol{x}$ and $\|\nabla f_i(\boldsymbol{x}) - \nabla f_i(\boldsymbol{y})\| \leq L \|\boldsymbol{x} - \boldsymbol{y}\|, \forall \boldsymbol{x}, \boldsymbol{y}, i$.

(A2) The mini-batch gradients in the algorithm are uniformly upper bounded, i.e. $\exists G \in \mathbb{R}$ s.t. $\|\boldsymbol{g}_{t,i}\| \leq G, \forall t, i$.

(A3) The objective function satisfies the *strong growth condition* with constant $r^2$, i.e. $\forall \boldsymbol{x}$, $\frac{1}{n} \sum_{i=1}^{n} \|\nabla f_i(\boldsymbol{x})\|^2 \leq r^2 \|\nabla f(\boldsymbol{x})\|^2$.

The strong growth condition assumption is essentially enforcing the norms of individual gradients to be at the same scale as the norm of the full gradient. This condition was originally used to derive a faster convergence for SGD in the context of convex finite sum minimization (Schmidt & Roux, 2013). It was further explored to analyze SGD-type methods after showing its closed relationship with interpolation (Ma et al., 2018; Vaswani et al., 2019; Gower et al., 2019). Built upon these previous analysis, we will give a more in-depth discussion for this assumption in section 6.

Under these assumptions, the following theorems show our convergence result for the full and diagonal versions of AdaGrad-window.

**Theorem 1** (The convergence rate of full AdaGrad-window). *For any $T > 4$, set $\eta = m^{-5/4}$, denote $C_1 = m^{5/4}\sqrt{11/3 + 8 \cdot r^2} \left(f(\boldsymbol{x}_{1,1}) - f^* + G\right)/\sqrt{2}$ and $C_2 = 5m^{5/4}\sqrt{11/3 + 8 \cdot r^2}L/\sqrt{2}$ as constants independent of $T$. We have:*

$$\min_{1 \leq t \leq T} \|\nabla f(\boldsymbol{x}_{t,1})\| \leq \frac{1}{\sqrt{T}}(C_1 + C_2 \ln T). \qquad (2)$$

**Theorem 2** (The convergence rate of diagonal AdaGrad-window). *For any $T > 4$, set $\eta = m^{-5/4}$, denote $C_1' = m^{5/4}\sqrt{11/3 + 8 \cdot r^2}\left(f(\boldsymbol{x}_{1,1}) - f^* + G\sqrt{d}\right)/\sqrt{2}$ and $C_2' = 5m^{5/4}\sqrt{11/3 + 8 \cdot r^2}d^{3/2}L/\sqrt{2}$ as constants independent of $T$. We have:*

$$\min_{1 \leq t \leq T} \|\nabla f(\boldsymbol{x}_{t,1})\| \leq \frac{1}{\sqrt{T}}(C_1' + C_2' \ln T). \qquad (3)$$

The interpretation of these two theorems is that we are able to find an approximate first-order stationary point such that $\|\nabla f(\boldsymbol{x})\| = \tilde{O}(T^{-1/2})$ within $T$ epochs using both versions. We notice that the convergence rate of AdaGrad-window matches that of GD when $m = 1$, which denotes that our results are relatively tight with respect to $T$. The complete proof is included in the appendix. We will give the intuition and key lemmas explaining how to utilize random shuffling and second moments to obtain these results in the next section.

In addition, we also prove for another variant of AdaGrad, namely, AdaGrad-truncation with second-moment matrix defined as $\boldsymbol{V}_{1,i} = m \cdot \boldsymbol{I}$ and $\boldsymbol{V}_{t,i} = m\|\sum_{j=1}^{m} \boldsymbol{g}_{t-1,j}\|^2 \cdot \boldsymbol{I}$ when $t > 1$ . This second-moment matrix is very similar to the norm version of AdaGrad (Ward et al., 2018) whereas we use the aggregation of mini-batch gradients in the previous epoch as the coefficient. AdaGrad-truncation is beneficial since the formulation leads to a fast and simple implementation without needing to discuss the full and diagonal versions. Due to the space limitation, we list the result below and defer the discussions to the appendix.

**Theorem 3** (The convergence rate of AdaGrad-truncation). *For any $T > 4$, set $\eta = \sqrt{3}/(10m^{1/2}Lr) \cdot \sqrt{f(\boldsymbol{x}_{1,1} - f^* + G^2)}/\sqrt{L+2}$, denote $C = 80m(Lr+2)\sqrt{f(\boldsymbol{x}_{1,1}) - f^* + G^2}$ as constants independent of $T$. We have:*

$$\min_{1 \leq t \leq T} \|\nabla f(\boldsymbol{x}_{t,1})\| \leq \frac{C}{\sqrt{T}}. \tag{4}$$

## 4 OVERVIEW OF ANALYSIS

The goal of this section is to give the key intuition for proving Theorem 1 and Theorem 2. In the following, we use the full version as an example where similar results can be obtained for the diagonal version by adding a dependency on dimension $d$. Proof details of both versions are included in the appendix. The key towards proving Theorem 1 is to establish the following lemma.

**Lemma 1.** *For any $t > 1$, in the full version of AdaGrad-window, denote $c_1 = \eta/\sqrt{11/3 + 8 \cdot r^2}, c_2 = 5\eta^2 m^2 L/2 + 5\eta^2 m^{5/2} L/\pi$ as constants independent of $t$. We have either:*

$$\frac{1}{\sqrt{t}} \cdot c_1 \|\nabla f(\boldsymbol{x}_{t,1})\| \leq f(\boldsymbol{x}_{t,1}) - f(\boldsymbol{x}_{t,m+1}) + \frac{1}{t} \cdot c_2, \tag{5}$$

*or $\|\nabla f(\boldsymbol{x}_{t,1})\| \leq \frac{1}{\sqrt{t}} \cdot (\eta m L)$. In addition, there is always $0 \leq f(\boldsymbol{x}_{t,1}) - f(\boldsymbol{x}_{t,m+1}) + \frac{1}{t} \cdot c_2$.*

The deduction of convergence rate is straightforward based on this lemma. Either there is $\|\nabla f(\boldsymbol{x}_{t,1})\| \leq (\eta m L/\sqrt{2})/\sqrt{T}$ for some $T/2 \leq t \leq T$, leading directly to Theorem 1, or we can sum up equation (5) for $T/2 \leq t \leq T$: the coefficients are approximately $\sqrt{T}$ on the left and $\ln T$ on the right thus leading to Theorem 1. Therefore, we turn to the proof of this lemma instead. Under the $L$-smooth assumption, we have the standard descent result for one epoch $\nabla f(\boldsymbol{x}_{t,1})^\top (\boldsymbol{x}_{t,1} - \boldsymbol{x}_{t,m+1}) \leq f(\boldsymbol{x}_{t,1}) - f(\boldsymbol{x}_{t,m+1}) + L/2 \cdot \|\boldsymbol{x}_{t,m+1} - \boldsymbol{x}_{t,1}\|^2$ (we refer the proof of to (Nesterov, 2018)). Rewrite the equation by replacing $\boldsymbol{x}_{t,m+1} - \boldsymbol{x}_{t,1}$ on the left with the AdaGrad-window updates:

$$\frac{\eta}{\sqrt{t}} \cdot \underbrace{\nabla f^\top(\boldsymbol{x}_{t,1})\boldsymbol{V}_{t,m}^{-1/2}(\sum_{i=1}^{m} \boldsymbol{g}_{t,i})}_{\text{S1}} \leq f(\boldsymbol{x}_{t,1}) - f(\boldsymbol{x}_{t,m+1}) + \frac{L}{2}\|\boldsymbol{x}_{t,m+1} - \boldsymbol{x}_{t,1}\|^2$$

$$+ \frac{\eta}{\sqrt{t}} \cdot \underbrace{\nabla f(\boldsymbol{x}_{t,1})^\top \left[\boldsymbol{V}_{t,m}^{-1/2}\sum_{i=1}^{m}\boldsymbol{g}_{t,i} - \sum_{i=1}^{m}\boldsymbol{V}_{t,i}^{-1/2}\boldsymbol{g}_{t,i}\right]}_{\text{S2}}.$$

The idea behind this decomposition is to split out the term S1 that behaves similarly to the full gradient and control the remaining terms. Similar to other analysis of adaptive methods, the term $\|\boldsymbol{x}_{t,m+1} - \boldsymbol{x}_{t,1}\|^2$ on the right can be upper bound by a constant times $1/t$ (we refer scalars that do not depend on $t$ to constants). This can be done by simply plugging in the update rules, the details of which are showed in the appendix. Next, we show how to bound term S1 and S2 in order to prove Lemma 1.

### 4.1 LOWER BOUND OF S1

To obtain the lower bound of S1, we need two steps. The first step stems from the idea in random shuffling SGD (Haochen & Sra, 2019), which is to bound the difference between the full gradient and the aggregation of mini-batch gradients. Formally, we have the following lemma.

**Lemma 2.** *For any $t > 0$, in the full version of AdaGrad-window, denote constant $c_3 = \eta(m-1)L/2$, we have:*

$$\|\nabla f(\boldsymbol{x}_{t,1}) - \frac{1}{m}\sum_{i=1}^{m}\boldsymbol{g}_{t,i}\| \leq \frac{1}{\sqrt{t}} \cdot c_3. \tag{6}$$

Building on top of this lemma, term S1 can be written into a constant combination of $1/\sqrt{t}$ and $(\sum_{i=1}^{m}\boldsymbol{g}_{t,i})^{\top}\boldsymbol{V}_{t,m}^{-1/2}(\sum_{i=1}^{m}\boldsymbol{g}_{t,i})$, which leads to our second step. In the second step, we utilize the strong growth assumption to obtain a lower bound formulated as below.

**Lemma 3.** *For any $t > 0$, in the full version of AdaGrad-window, we have either:*

$$(\sum_{i=1}^{m}\boldsymbol{g}_{t,i})^{\top}\boldsymbol{V}_{t,m}^{-1/2}(\sum_{i=1}^{m}\boldsymbol{g}_{t,i}) \geq \frac{1}{\sqrt{11/3 + 8 \cdot r^2}}\|\sum_{i=1}^{m}\boldsymbol{g}_{t,i}\|, \tag{7}$$

*or $\|\nabla f(\boldsymbol{x}_{t,1})\| \leq \frac{1}{\sqrt{t}} \cdot (\eta m L)$. In addition, there is always $(\sum_{i=1}^{m}\boldsymbol{g}_{t,i})^{\top}\boldsymbol{V}_{t,m}^{-1/2}(\sum_{i=1}^{m}\boldsymbol{g}_{t,i}) \geq 0$.*

This lemma shows that $(\sum_{i=1}^{m}\boldsymbol{g}_{t,i})^{\top}\boldsymbol{V}_{t,m}^{-1/2}(\sum_{i=1}^{m}\boldsymbol{g}_{t,i})$ can be lower bound by $\|\sum_{i=1}^{m}\boldsymbol{g}_{t,i}\|$ times a constant. Therefore, we are able to derive a constant combination of $\|\sum_{i=1}^{m}\boldsymbol{g}_{t,i}\|$ and $1/\sqrt{t}$ as the lower bound for S1, which is desired for Lemma 1.

We emphasize that the essential element of the convergence rate improvement lies in Lemma 3. For SGD, the matrix $\boldsymbol{V}_{t,m}$ is the identity matrix leading to a lower bound of $\|\sum_{i=1}^{m}\boldsymbol{g}_{t,i}\|^2$ instead of $\|\sum_{i=1}^{m}\boldsymbol{g}_{t,i}\|$. This lower order leads to a greater decrease of objective value between consecutive epochs as shown in Lemma 1. The reason that we are able to lower the order on $\|\sum_{i=1}^{m}\boldsymbol{g}_{t,i}\|$ is due to the presence of the second moments canceling out the order.

### 4.2 UPPER BOUND OF S2

To obtain the upper bound of S2, we can write $\boldsymbol{V}_{t,m}^{-1/2}\sum_{i=1}^{m}\boldsymbol{g}_{t,i}$ into $\sum_{i=1}^{m}\boldsymbol{V}_{t,m}^{-1/2}\boldsymbol{g}_{t,i}$. Therefore, we only need to take care of the second-moment matrices $\boldsymbol{V}_{t,m}$ and $\boldsymbol{V}_{t,i}$. As a matter of fact, we have the following lemma.

**Lemma 4.** *For any $t > 1$ and $1 \leq i \leq m$, in the full version of AdaGrad-window, denote $c_4 = 6\eta(m-i-1)(m-i)L/\pi + 4\eta(m-i)(m+i+1)L/\pi$ as constants independent of t, we have:*

$$\|\boldsymbol{V}_{t,m}^{1/2} - \boldsymbol{V}_{t,i}^{1/2}\| \leq \frac{1}{\sqrt{t}} \cdot c_4. \tag{8}$$

Based on this lemma, we can obtain an upper bound of S2 using the result below.

**Lemma 5.** *For any $t > 1$, in the full version of AdaGrad-window, denote $c_5 = 5\eta m^{5/2}/\pi L + \eta m^2 L$ as constants independent of t, we have:*

$$\nabla f(\boldsymbol{x}_{t,1})^{\top}\left[m\boldsymbol{V}_{t,m}^{-1/2}\nabla f(\boldsymbol{x}_{t,1}) - \sum_{i=1}^{m}\boldsymbol{V}_{t,i}^{-1/2}\boldsymbol{g}_{t,i}\right] \leq \frac{1}{\sqrt{t}} \cdot c_5. \tag{9}$$

With this upper bound derived, we are able to prove Lemma 1, which is the key intermediate result towards the convergence result in the Theorem 1.

## 5 COMPLEXITY ANALYSIS

Based on Theorem 1 and 2, we discuss the computational complexity for two versions of AdaGrad-window. We compare the total complexity between AdaGrad-window and random shuffling SGD to demonstrate that this adaptive gradient method can be faster than SGD after finite epochs in theory.

**Corollary 1** (The computational complexity of full-version AdaGrad-window). *Let $\{\boldsymbol{x}_{t,1}\}_{t=1}^T$ be the sequence generated by AdaGrad-window. For given tolerance $\epsilon$, to guarantee that $\min_{1 \le t \le T} \|\nabla f(\boldsymbol{x}_{t,1})\| \le \epsilon$, the total number of epochs is nearly (ignoring logarithm) $O(m^{5/2}\epsilon^{-2})$. Therefore, the total number of gradient evaluations is nearly (ignoring logarithm) $O\left(m^{5/2}n\epsilon^{-2}\right)$.*

Compared with the full version, their diagonal version is more practical in modern neural network training. Fortunately, a similar result can be derived for the diagonal version.

**Corollary 2** (The computational complexity of diagonal-version AdaGrad-window). *Let $\{\boldsymbol{x}_{t,1}\}_{t=1}^T$ be the sequence generated by AdaGrad-window. For given tolerance $\epsilon$, to guarantee $\min_{1 \le t \le T} \|\nabla f(\boldsymbol{x}_{t,1})\| \le \epsilon$, the total number of epochs is nearly (ignoring logarithm) $O(m^{5/2}d^3\epsilon^{-2})$. Therefore, the total number of gradient evaluations is nearly (ignoring logarithm) $O\left(m^{5/2}nd^3\epsilon^{-2}\right)$.*

For achieving the $\epsilon$-approximate first-order stationary point, the total number of gradient evaluation required by random shuffling SGD is $O(nd\epsilon^{-3})$ (Haochen & Sra, 2019; Nguyen et al., 2020). In a rough comparison, AdaGrad-window has advantages over random shuffling SGD when $\epsilon \approx O(m^{-5/2})$. Therefore, in theory, AdaGrad-window is more efficient when the number of iterations in one epoch $m$ is small. Recent works in deep neural net training (You et al., 2017; 2020) have shown that in large batch scenarios, adaptive methods tend to converge faster in training. Since $m$ is the number of iterations in one epoch, meaning that small $m$ gives a large batch size, our theory supports these previous findings. However, it seems that choosing $m = 1$, which amounts to full gradient calculation every step, attains the fastest result theoretically. We point out here that our bound on the $m$ might not be strict and encourage future work to improve upon that in order to get a sense of how to choose mini-batch sizes in practice.

## 6 THE STRONG GROWTH CONDITION ASSUMPTION

This section aims to provide answers regarding the strong growth condition assumption (A3). In particular, when is this assumption satisfied? Can it be used to improve the convergence rate of other methods? As answers to the first question, we first summarize previous results based on its closed relation with interpolation and the weak growth condition (Vaswani et al., 2019), then we extend to show that two general types of objective functions satisfy the strong growth condition in some cases. We discuss the convergence rate of other gradient-based methods under the strong growth condition for the second question.

### 6.1 FUNCTIONS SATISFYING THE STRONG GROWTH CONDITION ASSUMPTION

The strong growth condition in (A3) requires $\|\nabla f_i(\boldsymbol{x})\| = 0$ for all the individual functions at local stationary point $\boldsymbol{x}$ with $\|\nabla f(\boldsymbol{x})\| = 0$. In the sense of fitting model to data, where each $f_i$ represents a data point, this translates to $\boldsymbol{x}$, as the parameter, interpolating all the data points. The interpolation property has been utilize to study the convergence and mini-batch size of SGD for convex loss functions (Ma et al., 2018). Formally, function $f$ is said to satisfy the *interpolation property*, if for any $\boldsymbol{x}$:

$$\nabla f(\boldsymbol{x}) = 0 \Rightarrow \nabla f_i(\boldsymbol{x}) = 0, \forall i. \tag{10}$$

This property has been observed to hold for expressive models such as over-parameterized neural networks (Zhang et al., 2017).

Another concept closely related to the strong growth condition is the weak growth condition. Defined by Vaswani et al. (2019), function $f$ is said to satisfy the *weak growth condition* with constant $\rho$, if for any $\boldsymbol{x}$:

$$\frac{1}{n} \sum_{i=1}^n \|\nabla f_i(\boldsymbol{x})\|^2 \le 2\rho L[f(\boldsymbol{x}) - f(\boldsymbol{x}^*)], \tag{11}$$

where $L$ is the smoothness constant and $\boldsymbol{x}^*$ is a minima of $f$, assuming existence. The weak growth condition is a relaxed version of the strong growth condition in that the latter implies the former. Vaswani et al. (2019) also showed that functions satisfying both the weak growth condition and Polyak-Lojasiewicz (PL) inequality (Polyak, 1963) must satisfy the strong growth condition.

Furthermore, they proved that convex functions with interpolation property must satisfy the weak growth condition. The following diagram summarized the above results.

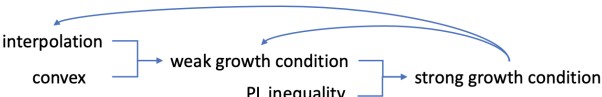

Figure 1: Previous results on the strong growth condition

Since the PL inequality holds for strongly convex functions, strongly convex functions with interpolation property must satisfy the strong growth condition. Results in (Milne, 2019) demonstrate that the quadratic loss function with $l_2$-penalty for a feed-forward neural network with ReLU activation is piecewise strongly convex, with domains of convexity determined by the network architecture, penalty factor, and data. Based on this fact, we may have the following lemma for the strong growth condition.

**Lemma 6.** *Let $f_i(\boldsymbol{x})$ be the $l_2$-penalized quadratic loss function at data point $(\boldsymbol{a}_i, o_i)$ for a feed-forward neural network with ReLU activation and scalar outputs, i.e.*

$$f_i(\boldsymbol{x}) = l_{\boldsymbol{x}}(\boldsymbol{a}_i, o_i) + \frac{\lambda}{2}\|\boldsymbol{x}\|^2,$$

*where $\boldsymbol{x}$ is the parameter of the neural network and $\lambda$ is the penalty factor. Define a nonempty open set containing $0$ to be:*

$$U = \{\boldsymbol{x} : f(\boldsymbol{x})^{1/2}\|\boldsymbol{x}\|^{H-1} \leq \frac{\lambda}{2\sqrt{2}H(H+1)c}\},$$

*where $H$ is the number of hidden layers and $c \geq \|\boldsymbol{a}_i\|, \forall i$. If $f_i$ has a uniform minimizer $\boldsymbol{x}^*$ for $i = 1, ..., n$ and $f$ is $L$-smooth with constant $L$, then almost all points in $U$ satisfy the strong growth condition with constant $\frac{2L}{\lambda}$, i.e. $\exists U' \subset U$ such that $U \backslash U'$ has no inner point and:*

$$\frac{1}{n}\sum_{i=1}^{n}\|\nabla f_i(\boldsymbol{x})\|^2 \leq \frac{2L}{\lambda}\|\nabla f(\boldsymbol{x})\|^2, \quad \forall \boldsymbol{x} \in U'.$$

Therefore, for feed-forward neural networks with ReLU activation, e.g., VGG networks (Simonyan & Zisserman, 2015), we are able to show that the strong growth condition holds for $\boldsymbol{x}$ in the neighborhood of $0$, if there exists a uniform minimizer for all $f_i$. As a result, we might have the strong growth condition for all $\boldsymbol{x}_{t,i}$ generated by the algorithm if the initialization and step sizes are appropriately chosen.

For linear models, we are able to derive much stronger results such that the strong growth condition holds for all $\boldsymbol{x}$ without any constraint on the global minimizer. In particular, Vaswani et al. (2019) showed that the squared-hinge loss, for linearly separable data with a margin $\tau$ and finite support $c$, satisfies the strong growth condition with constant $\frac{c^2}{\tau^2}$. We extend this result to cross-entropy loss and obtain the same result.

**Lemma 7.** *For linear separable data with margin $\tau$ and finite support of size $c$, the cross-entropy loss satisfies the strong growth condition with constant $\frac{c^2}{\tau^2}$.*

## 6.2 CONVERGENCE RATE UNDER THE STRONG GROWTH CONDITION ASSUMPTION

Under the strong growth condition assumption, previous results have shown that SGD can achieve a better convergence rate in many settings. In the context of expectation minimization and global optimality, SGD has linear convergence for strongly convex problems and sublinear $O(T^{-1})$ convergence for convex problems (Schmidt & Roux, 2013). For non-convex problems, Vaswani et al. (2019) proved that constant step-size SGD can obtain first-order stationary points in an $O(T^{-1/2})$ rate asymptotically. However, to the best of our knowledge, non-asymptotic results under the strong growth condition have yet to be explored. Our main theorem shows that adaptive gradient methods can achieve a non-asymptotic convergence rate of $\hat{O}(T^{-1/2})$.

# 7 EXPERIMENTS

To investigate the effect of adaptive step size and random shuffling, we compare the empirical performances of four different methods on MNIST and CIFAR-10 to show the acceleration effect. We include SGD and AdaGrad to confirm the existing phenomenon that adaptive step size accelerates the convergence of training. We also show results of the modified counterparts, SGD-shuffle and AdaGrad-window, to demonstrate the additional benefits of shuffling in training. Both adaptive methods are taken to be the more practical diagonal version.

For our first experiment, we compare the results of four methods for logistic regression on MNIST. To further examine the performance on non-convex problems, we train ResNet-18 (He et al., 2015) for the classification problem on CIFAR-10. To back up our theoretical results in the last section where the convergence on the minimum of gradients across epochs is established, we report the best train loss and best test accuracy up-to-current-epoch in figure 2. We can see that adaptive methods perform better than SGD methods at the end in both training and testing. For the first few epochs of ResNet-18 training, we argue that the comparison seems contrary to the theory because of the constant effect in the convergence rate where this effect dies out when the epoch number increases. We can also see that SGD-shuffle and AdaGrad-window exhibit better convergence than their counterparts in training. The details for the experiments are in the appendix.

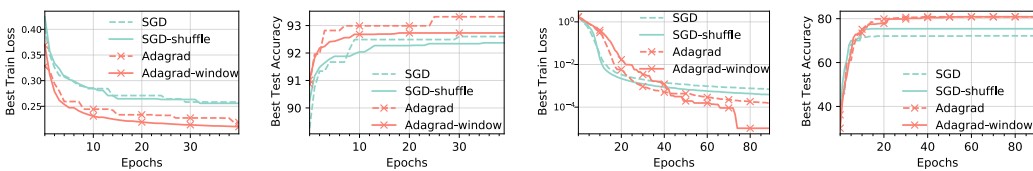

Figure 2: Left: best train loss and test accuracy up-to-current-epoch of logistic regression on MNIST. Right: best train loss and test accuracy up-to-current-epoch of ResNet-18 on CIFAR-10.

# 8 CONCLUSION

In this paper, we provide a novel analysis to demonstrate that adaptive gradient methods can be faster than SGD after finite epochs in non-convex and random shuffling settings. We prove that AdaGrad-window and AdaGrad-truncation obtain a convergence rate of $\tilde{O}(T^{-1/2})$ for first-order stationary points, a significant improvement compared with existing works. One key element is the strong growth condition, which is common in over-parameterized models. We also investigate the computational complexity and show that our theory supports recent findings on training with large batch sizes. We believe that this paper is a good start that could lead to analysis and practice in more general settings.

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
