# OpenReview forum: "Adaptive Gradient Methods Can Be Provably Faster than SGD with Random Shuffling"
_ICLR.cc/2021/Conference — Reject_

### Official Review · AnonReviewer3 · 2020-10-26
**Reviewer #3**

**Rating:** 4
**Confidence:** 4

**Review:**

I will initially provide a summary of the paper and list overall strengths and weakness of the paper. Then, I present my additional comments which are related to specific expressions in the main text, proof steps in the appendix etc. I would appreciate it very much if authors could address my questions/concerns under “Additional Comments” as well, since they affect my assessment and understanding of the paper; consequently my score for the paper.



Summary:

•	The paper focuses on convergence of two newly-proposed versions of AdaGrad, namely AdaGrad-window and AdaGrad-truncation, for finite sum setting where each component is smooth and possibly nonconvex.

•	The authors prove convergence rate with respect to number of epochs T, where in each epoch one full pass over the data is performed with respect to well-known “random shuffling” sampling strategy.

•	Specifically, AdaGrad-window is shown to achieve $\tilde{ \mathcal O } (T^{-1/2})$ rate of convergence, whereas AdaGrad-truncation attains $\mathcal( T^{-1/2} )$ convergence, under component-wise smoothness and bounded gradients assumptions. Additionally, authors introduce a new condition/assumption called consistency ratio which is an essential element of their analysis.

•	The paper explains the proposed modification to AdaGrad and provide their intuition for such adjustments. Then, the main results are presented followed by a proof sketch, which demonstrates the main steps of the theoretical approach.

•	In order to evaluate the practical performance of the modified adaptive methods in a comparative fashion, two set of experiments were provided: training logistic regression model on MNIST dataset and Resnet-18 model on CIFAR-10 dataset. In these experiments; SGD, SGD with random shuffling, AdaGrad and AdaGrad-window were compared. Additionally, authors plot the behavior of their proposed condition “consistency ratio” over epochs.



Strengths:

•	I think epoch-wise analysis, especially for finite sum settings, could help provide insights into behaviors of optimization algorithms. For instance, it may enable to further investigate effect of batch size or different sampling strategies with respect to progress of the algorithms after every full pass of data. This may also help with comparative analysis of deterministic and stochastic methods.

•	I have checked the proof of Theorem 1 in details and had a less detailed look at Theorems 2 and 3. I appreciate some of the technically rigorous sections of the analysis as the authors bring together analytical tools from different resources and re-prove certain results with respect to their adjustments.

•	Performance comparison in the paper is rather simple but the authors try to provide a perspective of their consistency condition through numerical evidence. It gives some rough idea about how to interpret this condition.

•	Main text is written in a clear; authors highlight their modification to AdaGrad and also highlight what their new “consistency condition” is. Proposed contributions of the paper are stated clearly although I do not totally agree with certain claims. One of the main theorems has a proof sketch which gives an overall idea about authors’ approach to proving the results.



Weaknesses:

•	Although numerically the paper provides an insight into the consistency condition, it is not verifiable ahead of time. One needs to run a simulation to get some idea about this condition, although it still wouldn’t verify the correctness. Since authors did not provide any theoretical motivation for their condition, I am not fully convinced out this assumption. For instance, authors could argue about a specific problem setting in which this condition holds.

•	Theorem 3 (Adagrad-truncation) sets the stepsize depends on knowledge of $r$. I couldn’t figure out how it is possible to compute the value $r$ ahead of time. Therefore, I do not think this selection is practically applicable. Although I appreciate the theoretical rigor that goes into proving Theorem 3, I believe the concerns about computing $r$ weakens the importance of this result. If I am missing out some important point, I would like to kindly ask the authors to clarify it for me.

•	The related work which is listed in Table 1, within the group “Adaptive Gradient Methods” prove \emph{iteration-wise} convergence rates for variants of Adam and AdaGrad, which I would call the usual practice. This paper argues about \emph{epoch-wise} convergence. The authors claim improvement over those prior papers although the convergence rate quantifications are not based on the same grounds. All of those methods consider the more general expectation minimization setting. I would suggest the authors to make this distinction clear and highlight iteration complexities of such methods while comparing previous results with theirs. In my opinion, total complexity comparison is more important that rate comparison for the setting that this paper considers.

•	As a follow up to the previous comment, the related work could have highlighted related results in finite sum setting. Total complexity comparisons with respect to finite sum setting is also important. There exists results for finite-sum nonconvex optimization with variance reduction, e.g., Stochastic Variance Reduction for Nonconvex Optimization, 2016, Reddi et. al. I believe it is important to comparatively evaluate the results of this paper with that of such prior work.

•	Numerically, authors only compare against AdaGrad and SGD. I would say this paper is a rather theory paper, but it claims rate improvements, for which I previously stated my doubts. Therefore, I would expect comparisons against other methods as well, which is of interest to ICLR community in my opinion.

•	This is a minor comment that should be easy to address. For ICLR, supplementary material is not mandatory to check, however, this is a rather theoretical paper and the correctness/clarity of proofs is important. I would say authors could have explained some of the steps of their proof in a more open way. There are some crucial expressions which were obtained without enough explanations. Please refer to my additional comments in the following part.



Additional Comments:

•	I haven’t seen the definition that $x_{t, m+1} = x_{t+1, 1}$ in the main text. It appears in the supplements. Could you please highlight this in the main text as it is important for indexing in the analysis?

•	Second bullet point of your contributions claim that “[consistency] condition is easy to verify”. I do not agree with this as I cannot see how someone could guarantee/compute the value $r$ ahead of time or even after observing any sequence of gradients. Could you please clearly define what verification means in this context?

•	In Assumption A3, I understand that $G_t e_i = g_{t,i}$ and $G_t e = \sum_{i=1}^{m} g_{t,i}$. I believe the existing notation makes it complicated for the reader to understand the implications of this condition.

•	In the paragraph right above Section 4.2, authors state that presence of second moments, $V_{t,i}$ enables adaptive methods to have improved rates of SGD through Lemma 3. Could the authors please explain this in details?

•	In Corollary 1, authors state that “the computational complexity is nearly $\tilde{ \mathcal O (m^{5/2}nd^2\epsilon^{-2}) }$”. A similar statement exists in Corollary 2. Could you please explain what “nearly” means in this context?

•	In Lemma 8 in the supplements, $aa^T$ and $bb^T$ in the main expression of the lemma are rank-1 matrices. This lemma has been used in the proof of Lemma 4. As far as I understood, Lemma 8 is used in such a way that $aa^T$ or $bb^T$ correspond to something like $g_{t,j}^2 – g_{t-1, j}^2$. I am not sure if this construction fits into Lemma 8 because, for instance, the expression $g_{t,j}^2 – g_{t-1, j}^2$ is difference of two rank-1 matrices, which could have rank \leq 2. Hence, there may not exist some vector $a$ such that $aa^T = g_{t,j}^2 – g_{t-1, j}^2$, hence Lemma 8 may not be applied. If I am mistaken in my judgment I am 100% open for a discussion with the authors.

•	In the supplements, in section “A.1.7 PROOF OF MAIN THEOREM 1”, in the expression following the first line, I didn’t understand how you obtained the last upper bound to $\| \nabla f(x_{t,i}) \|$. Could you please explain how this is obtained?



Score:

I would like to vote for rejecting the paper. I praise the analytically rigorous proofs for the main theorems and the use of a range of tools for proving the key lemmas. Epoch-wise analysis for stochastic methods could provide insight into behavior of algorithms, especially with respect to real-life experimental setting. However, I have some concerns:

1.	I am not convinced about the importance of consistency ratio and that it is a verifiable condition.

2.	Related work in Table 1 has iteration-wise convergence in the general expectation-minimization setting whereas this paper considers finite sum structure with epoch-wise convergence rates. The comparison with related work is not sufficient/convincing in this perspective.

3.	(Minor) I would suggest the authors to have a more comprehensive experimental study with comparisons against multiple adaptive/stochastic optimizers. More experimental insight might be better for demonstrating consistency ratio.



Overall, due to the reasons and concerns stated in my review, I vote for rejecting this paper. I am open for further discussions with the authors regarding my comments and their future clarifications.

======================================= Post-Discussions =======================================

I would like to thank the authors for their clarifications. After exchanging several responses with the authors and regarding other reviews, I decide to keep my score.

1. Although the authors come up with a more meaningful assumption, i.e., SGC, compared to their initial condition, I am not fully convinced about the contributions with respect to prior work: SGC assumption is a major factor in the improved rates and it is a very restrictive assumption to make in practice.

2. Although this paper proposes theoretical contributions regarding adaptive gradient methods, the experiments could have been a bit more detailed. I am not sure whether the experimental setup fully displays improvements of the proposed variants of AdaGrad.

---

> ### Author Response · Authors · 2020-11-20
> **Response to reviewer #3**
>
> We first want to thank the reviewer for the detailed and valuable comment. All the suggestions are helpful and very much appreciated. We would like to point that we upload a revision that has a summary in the general response. As for the listed questions in the review, we will give corresponding responses in the following.
>
> Weakness:
> - We change the consistency ratio assumption to the strong growth condition, which is very similar in form but with much nicer theoretical properties. The strong growth condition can be verifiable ahead of time without simulations, and we add in proofs to show that the assumption holds for two specific types of objective functions.
> - For the $r$ presented in Theorem 3, we agree this makes the theory a little less elegant. However many optimizers depend on unknown constant, e.g., L-smooth constant for gradient descent, and it will not harm the practicality since we usually tuning the constant in the step size.
> - For questions regarding epoch-wise comparisons. Thanks for pointing this out! It needs a bit of clarification. In the revision, we change the epoch-wise comparison to iteration-wise comparison. We also state the distinction between expectation minimization and finite sum minimization. As a result, non-asymptotic convergences for shuffling in the finite sum minimization would introduce a dependency on sample size $n$.
> - Following the response to the previous comment, in the clarification we made about finite sum minimization vs expectation minimization, we discuss briefly how various methods can reduce variance. However, we consider SVRG to be a distinctive technique. So to make the objective of this submission clear, which is to compare adaptive gradient methods vs SGD with random shuffling, we did not add quantitative comparisons with SVRG.
> - Numerically, we only compare AdaGrad and SGD (with/without shuffling) since that is the main objective of this submission. We hope that the revised version would answer questions regarding rate improvements and that the presented experiments can provide support to the main objective. In the future, with time less limited, we would add in comparisons which could be of interest.
>
> Additional comments:
> - Thanks for suggesting to put $x_{t+1,1}=x_{t,m+1}$ in the main text. We highlighted this in the main text where AdaGrad-type methods are introduced.
> - (2 & 3 bullet points) Please refer to the first answer in weakness. Thanks for the detailed doubts regarding the assumption. They helped a lot in our development of the revision.
> - In the large equation in section 4 with terms S1 and S2, the adaptive step size allows us to bound S1 using $\|\sum_{i=1}^m g_{t,i}\|$ instead of $\|\sum_{i=1}^m g_{t,i}\|^2$, which is the case for SGD ($V_{t,m}=I$).
> - We revise this in the new version. "nearly" is ignoring the logarithm terms.
> - We add in an explanation of how to apply Lemma 8 (Lemma 10 now) to Lemma 4. Lemma 8 is used in the first aligned equation of proofs for Lemma 4, where in the second inequality, we use Lemma 8 $m-i$ times by setting $C=\sum_{j=1}^{i+k} g_{t,j}^2+\sum_{j=0}^{m-i-2-k}g_{t-1,m-j}^2$ and $a=g_{t-1,i+k+1}, b = g_{t,i+k+1}$ (Both $a$ and $b$ are rank-1 vectors).
> - We add in a line of equation of how to bound $||\nabla f(x_{t,1})||$, it is essentially using triangle inequality and bounds on $||\nabla f(x_{t,1})-\frac{1}{m}\sum_{i=1}^m g_{t,i}||$ and $||\frac{1}{m}\sum_{i=1}^m g_{t,i}||$.

---

> > ### Comment · AnonReviewer3 · 2020-11-24
> > **Thank you for your responses to my initial review**
> >
> > I will respond to some of the individual comments and also make some general statements regarding your modifications:
> >
> > -	_We change the consistency ratio assumption to the strong growth condition, which is very similar in form but with much nicer theoretical properties. The strong growth condition can be verifiable ahead of time without simulations, and we add in proofs to show that the assumption holds for two specific types of objective functions._
> >
> > I think replacing your previous unverifiable condition with strong growth condition (SGC) makes your theoretical claims stronger. I must note that SGC itself is a very strong assumption to make and in most applications with real data, this condition does not hold. You explain when this condition holds in section 6, and it is evident from some of those explanations that you need strong assumptions on the data, initialization or the loss function to make sure SGC is satisfied.
> >
> > -	_For the r presented in Theorem 3, we agree this makes the theory a little less elegant. However many optimizers depend on unknown constant, e.g., L-smooth constant for gradient descent, and it will not harm the practicality since we usually tuning the constant in the step size._
> >
> > My comment about $r$ was related to your previous condition which was not verifiable ahead of time. For SGC, to the best of my knowledge, you need to know the parameter $r$ to show convergence. I don’t know of any result that adapts to $r$.
> >
> > -	Thank you for your other responses for my comments under Weaknesses.
> >
> > -	I would also like to thank for your responses for my comments Additional Comments. I will have a specific answer for one of those responses: _We add in an explanation of how to apply Lemma 8 (Lemma 10 now) to Lemma 4. Lemma 8 is used in the first aligned equation of proofs for Lemma 4, where in the second inequality, ..._
> >
> > I am assuming you are referring to rank-1 matrices $aa^T$ and $bb^T$ by saying "Both $a$ and $b$ are rank-1 vectors".
> >
> > Let’s focus on the construction proposed in the proof of Lemma 4. You set $C = \sum_{j=1}^{i+k} g_{t,j}^2 + \sum_{j=0}^{m-i-2-k} g_{t-1,m-j}^2$. However, if you look at the expression for $V_{t,m}^{1/2} = \delta \mathbb I_d + \sum_{j=1}^{m} g_{t,j}^2$, it does not have any gradient components from previous iteration (iteration t-1) such as $g_{t-1, j}$. Hence, I don’t think your construction of C is correct. In fact, for arbitrary $i$, I don’t see how you would make use of Lemma 10 by representing $V_{t,i}^{1/2} – V_{t,m}^{1/2}$ in the form of $(\delta \mathbb I_d + C + aa^T)^{1/2} – (\delta \mathbb I_d + C + bb^T)^{1/2}$.
> >
> > **Additional Comments based on the updated manuscript:**
> >
> > -	You claim that Vaswani et al., 2019, shows **asymptotic** convergence rate for SGD. However, their Theorem 3 seems to prove non-asymptotic convergence of SGD with a rate of $\mathcal O (1 / T^{1/2})$ for non-convex losses satisfying SGC.
> >
> > -	I don’t agree with the statement that “The key towards our convergence rate improvement is twofold: the epoch-wise analysis of random shuffling enables us to leverage the benefit of full gradients; the adaptive learning rates endow a better improvement of objective value in consecutive epochs.” I would say SGC is the key assumption that enables the fast rate of $\mathcal O (1/  T^{1/2})$ as there are known theoretical bounds for smooth nonconvex problems which cannot be improved without further/stronger assumptions. Would the authors agree with me on this statement? If I am missing out something, please correct my mistake.
> >
> > I am open for further discussions to make sure we understand each other's perspectives.

---

> > > ### Author Response · Authors · 2020-11-25
> > > **Thank you so much for your further responses**
> > >
> > > We will respond to your comments and answer your questions as follows:
> > >
> > > - _I think replacing your previous unverifiable condition with strong growth condition (SGC) makes your theoretical claims stronger. I must note that SGC itself is a very strong assumption to make and in most applications with real data, this condition does not hold. You explain when this condition holds in section 6, and it is evident from some of those explanations that you need strong assumptions on the data, initialization or the loss function to make sure SGC is satisfied._
> > >
> > > Thank you for your recognition of our revision. As stated in the first work that defines the strong growth condition, we admit that it is a very strong assumption in that it requires the model to interpolate the data. Although this interpolation property has been observed to hold for many overparameterized models with strong expressive ability, it would still require some additional assumptions in order to satisfy SGC, e.g. weak growth condition + PL inequality. What we've shown in section 6 is that, without these kinds of additional assumptions, SGC is satisfied by two types of models (one of which SGC is satisfied by a constraint set of $x$). However, it remains unsolved that SGC can be satisfied by general models. Some partial answers to this might be the relaxations of SGC, e.g. weak growth condition [1] or expected smoothness [2].
> > >
> > > - _My comment about r was related to your previous condition which was not verifiable ahead of time. For SGC, to the best of my knowledge, you need to know the parameter r to show convergence. I don’t know of any result that adapts to r._
> > >
> > > Thank you for the clarification of the question. However, we are still not sure if the previous answer would adapt to this question. Our response regarding theorem 3 is that here it needs $r$ to show convergence like many other results that need $L$ (the L-smoothness constant) to show convergence. These parameters might not be given to us beforehand. But in practice, since these parameters appear in step size and we usually determine the stepsize by tuning it, it doesn't have much impact on experiments.
> > >
> > > - _I am assuming you are referring to rank-1 matrices $aa^T$ and $bb^T$  by saying "Both $a$ and $b$ are rank-1 vectors". Let’s focus on the construction proposed in the proof of Lemma 4. You set_ $C=\sum_{j=1}^{i+k} g_{t,j}^2+\sum_{j=0}^{m-i-2-k} g_{t-1,m-j}^2$. _However, if you look at the expression for_ $V_{t,m}^{1/2}=\delta I_d+\sum_{j=1}^{m}g_{t,j}^2$, _it does not have any gradient components from previous iteration (iteration t-1) such as_ $g_{t−1,j}$. _Hence, I don’t think your construction of $C$ is correct. In fact, for arbitrary $i$, I don’t see how you would make use of Lemma 10 by representing_ $V_{t,i}^{1/2}–V_{t,m}^{1/2}$ _in the form of_ $(\delta I_d+C+aa^T)^{1/2}-(\delta I_d+C+bb^T)$.
> > >
> > > To answer this question, first let's reconfirm the definitions:
> > >
> > > $$ V_{t,m} = \delta I_d+\sum_{j=1}^{m}g_{t,j}^2,\quad V_{t,i} = \delta I_d+\sum_{j=1}^{i}g_{t,j}^2+\sum_{j=i+1}^{m}g_{t-1,j}^2 $$
> > >
> > > Next, in order to use Lemma 10, we represents $V_{t,i}^{1/2}-V_{t,m}^{1/2}$ in the form of
> > > $$
> > > \sum_{k=0}^{m-i-1} \big[(\delta I_d+C_k+a_ka_k^T)^{1/2} - (\delta I_d+C_k+b_kb_k^T)^{1/2}\big]
> > > $$
> > > instead of $(\delta I_d+C+aa^T)^{1/2}-(\delta I_d+C+bb^T)$. Therefore, we actually use Lemma 10 for **m-i times** instead of one time.
> > >
> > > The way to construct $C_k,a_k,b_k$ so that $V_{t,i}^{1/2}-V_{t,m}^{1/2}=\sum_{k=0}^{m-i-1} \big[(\delta I_d+C_k+a_ka_k^T)^{1/2} - (\delta I_d+C_k+b_kb_k^T)^{1/2}\big]$ is by setting
> > > $$
> > > C_k = \sum_{j=1}^{i+k} g_{t,j}^2 + \sum_{j=0}^{m-i-2-k}g_{t-1,m-j}^2,\quad a_k = g_{t-1,i+k+1},\quad b_k=g_{t,i+k+1}
> > > $$
> > > so that $-(\delta I_d+C_k+b_kb_k^T)^{1/2}$ and $(\delta I_d+C_{k+1}+a_{k+1}a_{k+1}^T)^{1/2}$ cancel out, $(\delta I_d+C_{0}+a_{0}a_{0}^T)^{1/2}=V_{t,i}^{1/2}$ and $(\delta I_d+C_{m-i-1}+b_{m-i-1}b_{m-i-1}^T)^{1/2}=V_{t,m}^{1/2}$ (note that $\sum_{j=0}^{-1}$ is considered to be $0$).
> > >
> > > Does this answer your question?

---

> > > > ### Author Response · Authors · 2020-11-25
> > > > **Additional responses due to space limits**
> > > >
> > > > - _You claim that Vaswani et al., 2019, shows **asymptotic** convergence rate for SGD. However, their Theorem 3 seems to prove non-asymptotic convergence of SGD with a rate of_ $O(1/T^{1/2})$ _for non-convex losses satisfying SGC._
> > > >
> > > > Their theorem 3 is asymptotic where they showed $\min_{i} \mathbb{E} [\|\nabla f(w_i)\|]\leq \frac{2\rho L}{k} [f(w_0)-f^*]$. The expectation is taken with respect to the stochasticity in gradients. This expectation actually helps a lot in proving of theorem 3 as they took expectation with respect to $z_0,...,z_{t-1}$ in appendix B.2.
> > > >
> > > > - _I don’t agree with the statement that “The key towards our convergence rate improvement is twofold: the epoch-wise analysis of random shuffling enables us to leverage the benefit of full gradients; the adaptive learning rates endow a better improvement of objective value in consecutive epochs.” I would say SGC is the key assumption that enables the fast rate of $O(1/T^{1/2})$ as there are known theoretical bounds for smooth nonconvex problems which cannot be improved without further/stronger assumptions. Would the authors agree with me on this statement? If I am missing out something, please correct my mistake._
> > > >
> > > > Thank you so much for asserting your concerns. We would say that the statement regarding our convergence rate improvement is still correct, as one can see from the overview of analysis in section 4. Undoubtedly, SGC plays a key part in this result. However, we think that this statement includes SGC as "the adaptive learning rates endow a better improvement of objective value in consecutive epochs" uses SGC to show a better improvement. Maybe to make this statement more clear, we should modify it into **“The key towards our convergence rate improvement is twofold: the epoch-wise analysis of random shuffling enables us to leverage the benefit of full gradients; the adaptive learning rates and the strong growth condition endow a better improvement of objective value in consecutive epochs.”**
> > > >
> > > > [1] Sharan Vaswani, Francis Bach, and Mark Schmidt. Fast and faster convergence of sgd for over- parameterized models and an accelerated perceptron, 2019.
> > > >
> > > > [2] Robert Mansel Gower, Nicolas Loizou, Xun Qian, Alibek Sailanbayev, Egor Shulgin, and Peter Richtarik. Sgd: General analysis and improved rates, 2019.

---

> > > > > ### Comment · AnonReviewer3 · 2020-11-25
> > > > > **Answering your last part of the responses**
> > > > >
> > > > > I think there is a misunderstanding of the results of Vaswani et al., 2019. Could you please explain why you are saying this rate is asymptotic? To my understanding, the rate is non-asymptotic as the attainment of the rate does not require any quantity to approach zero in the limit.
> > > > >
> > > > > The proof for their Theorem 3 is pretty standard: they cannot evaluate the full expectation on its own with the set of assumptions they have as any iterate depends on the stochasticity in all the previous iterations (which is introduced through stochastic gradients). Hence, one needs to use some properties of expectation. Let $X$ be an $F$-measurable random variable and let $F_1 \subset F_2 \subset ... \subset F_n \subset F$ be a sequence of sub-sigma algebras of $F$. Then, $\mathbb E ( X ) = \mathbb E( \mathbb E(X \mid F_n) )$. Moreover, $\mathbb E( \mathbb E( X \mid F_1) \mid F_2 ) = \mathbb E( \mathbb E( X \mid F_2) \mid F_1 ) = \mathbb E( X \mid F_1 )$ almost surely. Their proof basically uses a combination of these properties which is standard in analysis of SGD-type methods. In fact, the first property is a special case of the second.
> > > > >
> > > > > After having a second look at your analysis, I got confused about the fact that you are not dealing with stochasticity of your gradients, $g_{t,i}$. The randomness in your setting is due to partitioning of the dataset into batches $[ \mathbb B_i ]_{i = 1, ..., m}$, but you are not taking this randomness into account. I am yet to figure the reason for this out. Could you please explain to me if I am missing any point?
> > > > >
> > > > > As always, I am open for further exchange of comments.

---

> > > > > > ### Author Response · Authors · 2020-11-25
> > > > > > **Thank you for your quick response**
> > > > > >
> > > > > > By saying asymptotic, we meant that the results of Vaswani et al., 2019 require to take expectation over the randomness in gradients, where you point out that they used $\mathbb{E}(X|F_n) = \mathbb{E}(X)$ ($X$ is the stochastic gradient in iteration $n$ to my understanding) like standard analysis of SGD. Sorry if the terminology of _asymptotic_ used here is confusing.
> > > > > >
> > > > > > In our analysis (which is to study the algorithm under random shuffling), the randomness comes from the partitioning of datasets. One can achieve results without taking the expectation over this randomness in random shuffling because the epoch-wise analysis can cancel out the effect of randomness. See, for example, Theorem 4 in (Nguyen et al., 2020). An intuitive understanding could be that in each epoch, we iterate through the entire dataset so that we do not need to account for this partitioning if we analyze epoch by epoch. Does this provide some answer to your question?
> > > > > >
> > > > > >
> > > > > >
> > > > > > [1] Lam M Nguyen, Quoc Tran-Dinh, Dzung T Phan, Phuong Ha Nguyen, and Marten van Dijk. A unified convergence analysis for shuffling-type gradient methods. arXiv preprint arXiv:2002.08246, 2020.

---

> > > > > > > ### Comment · AnonReviewer3 · 2020-11-25
> > > > > > > **follow up comments to convergence rate discussions**
> > > > > > >
> > > > > > > I think your definition of asymptotic convergence should be explained because from the perspective of optimization terminology, asymptotic convergence rate means something else and creates a confusion.
> > > > > > >
> > > > > > > Basically, you are relying on epoch-wise analysis, as adopted in Nguyen et al., 2020, but this leads to an additional $\sqrt{n}$ factor in your convergence rate. I understand your point about epoch-wise convergence quantification.
> > > > > > >
> > > > > > > Thank you for the quick response.

---

> > > > > > > > ### Author Response · Authors · 2020-11-25
> > > > > > > > **further replies**
> > > > > > > >
> > > > > > > > Thank you for your quick response. We upload a minor revision based on your comments on the definition of asymptotic convergence and statements of convergence rate improvement.

---

> > > > ### Comment · AnonReviewer3 · 2020-11-25
> > > > **thank you for clarifying proof of Lemma 4**
> > > >
> > > > I tried to verify this derivation for Lemma 4 and it seems correct to me. Thank you for the detailed answer.

---

### Official Review · AnonReviewer1 · 2020-10-26
**Adaptive Gradient Methods Can Be Provably Faster than SGD with Random Shuffling**

**Rating:** 4
**Confidence:** 4

**Review:**

This paper develops new stepsize rules for Adagrad with shuffling to improve the convergence rate from $O(1/T^{1/3})$ in the previous works to $\tilde{O}(1/T^{1/2})$, which is significantly better than existing AdaGrad variants.
However, after reading up to the analysis as well as the assumptions stated in the paper, it appears that Assumption 3 seems to be artificially enforced to achieve such a fast rate. Unfortunately, the authors do not provide examples or mathematical justification to show why this assumption is reasonable. This assumption is not algorithmic independent, it depends on the sequences generated by the algorithm. Although the authors conduct some experiments to show that this assumption holds in practice, but such an example remains nonconstructive and may hold by accident.

Though the proof technique seems to be new, it heavily relies on Assumption 3 to transform the squared-norm of the gradient as usually used in gradient-based methods for nonconvex problems to the norm of the gradient, making the convergence rate to be faster than known results as can be seen in the key recursive inequality (5) of Lemma 1.

In terms of technical details, the reviewer would like to raise the following questions?
1) What should be the nature of Assumption A.3.?
2) How should it relate to existing assumptions, e.g., strong convexity-type or bounded variance condition?
3) Is there any example that is independent of algorithms such that this assumption can be twisted to satisfy?
4) This rate seems to match the rate of SGD in the strongly convex case? Is there any relation between Assumption A3. and the strong convexity? If so, what is the meaning of the constant r?
5) Can the stepsize in Theorems  2 and 3 be independent of $f^*$?
6) Can one choose constant stepsize $\eta_t = \eta/T$ so that the $\log(T)$ term in the convergence rate disappear?

Since the main contribution is on the new choice of step-size/learning rate, it is very hard to assess the work from a practical point of view if one just simply tunes the learning rate in standard Adagrad to compete with the new algorithm. So, I do not see the experiment reflects much the efficiency of the new algorithm if the test is only done with NN training. Therefore, additional examples could be added to illustrate the benefit of the new variant.

In addition, the paper still contains so many typos as well as some inconsistent English expressions. Here are some of them: bouned,
intuitions, we refer scalars that does not depend on t to constants, ormance, seems off, etc.

---

> ### Author Response · Authors · 2020-11-20
> **Response to reviewer #1**
>
> Thank you so much for the valuable comment and questions! We upload a revised version of our paper, in which the consistency ratio assumption is replaced by the strong growth assumption. These two assumptions are very similar in form but the latter has much nicer property and could answer a lot of the related questions. Please refer to the general response where we summarize the difference between the two versions.
>
> For the technical questions, we answer them as follows:
> 1. It is very similar to the strong growth condition assumption in form, where the main proof is almost intact after changing A3. The strong growth condition assumption is enforcing the norms of individual gradients to be bounded by the norm of the full gradient and has close relation with interpolation. The details are discussed in section 6.1.
> 2. As following the previous response, we would answer questions in the context of the strong growth condition assumption. In our revision, we summarize the relationship between the strong growth condition and other assumptions in section 6.1.
> 3. In section 6.1, we show that the assumption holds for two specific types of objective functions.
> 4. Following the answer to the second question, strong convexity plus interpolation implies Assumption 3, where $r$ is closely related to the strongly convex constant and L-smooth constant. This can be seen from the proof of Lemma 6. SGD under assumption 3 and strongly convexity actually converges linearly. We discuss this in section 6.2.
> 5. The step size in theorem 2 is independent of $f^*$. As for theorem 3, we agree dependencies on $f^*$ makes it a little less elegant. However many optimizers depend on unknown constant, e.g., L-smooth constant for gradient descent, and it will not harm the practicality since we usually tuning the constant in the step size.
> 6. If we choose $\eta_t = \eta/T$, we would have a much worse convergence rate of $O(1/\ln T)$, since the term on the left side of lemma 1 would add up to $\ln T$.
>
> Our contribution is mainly on the theoretical analysis, as stated in the introduction. Numerically, we only compare AdaGrad and SGD (with/without shuffling) since that is the main objective of this submission and the presented experiments can provide support to the main objective. We apologize for the typos and change them in the revision.

---

### Official Review · AnonReviewer2 · 2020-10-28
**nice results, could do with more discussion of the assumptions.**

**Rating:** 7
**Confidence:** 4

**Review:**

This paper shows that adaptive learning rates are beneficial for finding critical points of finite-sum optimization problems. In particular, with appropriate learning rates, a variant of adagrad can find a epsilon critical point in \tilde O(1/epsilon^2) iterations. This improves upon previous results of either O(1/\epsilon^3) or O(1/\epsion^4) in various situations. The key new assumption is a “consistency condition” that bounds how big individual example gradients can be when the overall gradient is small.

Overall I liked this work, the result seems interesting, and potentially will inspire future work.

I liked the consistency condition here, but I think the paper would be very much served by a more in-depth discussion of what this condition is saying. My own intuition here is that, roughly speaking, the standard deviation of the gradients is at most r sqrt(m) larger than the true average gradient for some constant r.
This seems like some kind of analog of the “L*” bounds in convex stochastic optimization, for which it is known that when the gradients have variance at the optimum (which also implies small loss values at the optimum in the convex case), then asymptotically faster convergence is possible. Moreover, in this case it is also known that adaptive methods like adagrad are able to achieve these rates easily. See, e.g. (https://parameterfree.com/2019/09/20/adaptive-algorithms-l-bounds-and-adagrad/).

My main qualm with the paper is that the modified algorithms appear to require O(m) memory in order to implement the windowed adagrad. This seems a bit excessive, but seems a good direction for future work. For example, it is possible that some kind of EMA would be able to remove this issue.

However, related to this issue, it seems that the overall computational complexity of the algorithm is actually *worse* than just plain full-batch gradient descent, which would not require any memory overhead and takes O(n/epsilon^2) gradient evaluations to reach an epsilon critical point, rather than O(m^(5/2)n/epsilon^2) so it is unclear that this analysis really explains any success over full-batch gradient descent. Is it possible that the m factor is very loose in this analysis, or do I miss something here?

For Lemma 6 something seems a little off in proving the last statement. I believe the result is correct, but something is perhaps missing in the second equality, or at the very least could do with more explanation. For example, if d=1, m=2, \delta=0 and g_1 =1 and g_2=-1, then the relevant value is zero, but the second equality would say it is sqrt(2).

---

> ### Author Response · Authors · 2020-11-20
> **Response to reviewer #2**
>
> Thank you so much for the sharing of thoughts and doubts! As responses to the several points in the review, we answer them in the following.
>
> - (In-depth discussions about the assumption) Indeed, as we recently discovered, the consistency condition here is very similar to the strong growth condition, which essentially needs to bound the stochastic gradients using the true gradient. It has a close relationship with other convex assumptions. Under this assumption, several asymptotic better convergences have been shown. We discuss this in section 6 of our revision.
>
> - (Doubts about memory) Thanks for pointing this out! Roughly speaking, the update is very similar to the exponential moving average since it only focuses on recent gradients. We suspect that the analysis could be applied to other methods, such as Adam, RMSprop, AMSGrad, etc.
>
> - (Comparison with GD) Yes, it seems that this would be an issue for the non-convex setting where SGD and AdaGrad can not beat GD in theory. Lowering the order on m could be potential future improvements. It would also help provide some insights into how to choose the size of mini-batches.
>
> - (Details of proof) In the proof of Lemma 6 (Lemma 8 now), the second equality is actually an inequality with $\leq$ on the bottom row. Therefore, it doesn't violate the example.

---

### Official Review · AnonReviewer4 · 2020-10-29
**No benefit over full GD**

**Rating:** 3
**Confidence:** 4

**Review:**

*Summary:
This paper investigates stochastic methods for finding an approximate stationary point for a non-convex function that can be written as a finite sum. The authors consider the combination of adaptive (Adagrad style) methods in conjunction with a random shuffling.

*Significance:
The authors have missed a very important aspect of stochastic optimization.
In stochastic optimization the right way to measure progress is to present the error versus the number of stochastic gradient computations.
The authors present the error as a function of the number of epochs, where in each epoch we go over the whole dataset.

Moreover, the suggested method does not have any benefit over full GD.
Basically, when m=1, the method in the paper is equivalent to GD and obtains the same optimal
$O(1/\sqrt{T})$ rate. When $m>1$ the authors prove a rate which is worse by a factor of $O(m^{5/4})$ compared to GD. Therefore, they actually show that GD (m=1) is optimal for their algorithm and that there is no benefit to stochasticity (m > 1).


*Summary of review:
The suggested method  does not show any benefit over full GD, so I don't see what is the contribution here.

---

> ### Author Response · Authors · 2020-11-20
> **Response to reviewer #4**
>
> Thanks for the review. However, we are very sorry about the comment. We think that the reviewer has missed out on quite a few major parts of our submission and would like to respond by making several clarifications about the review.
>
> First of all, in both versions, we measure our methods in terms of gradient evaluations. We apologize that we did not highlight this in the introduction and change this in the revision.
>
> Secondly, the reviewer pointed out that the method studied in this paper is equivalent to GD when m=1. This is not correct, since like the original AdaGrad, we would still have an adaptive step size of $\eta (\delta I+ g_{t,1}g_{t,1}^\top)^{-1}$ when m=1. Furthermore, it is a general issue for the non-convex setting where SGD and AdaGrad can not beat GD in theory. What we have proved is that adaptive gradient methods can achieve the same order on $T$ as GD. Although the complexity analysis might seem that choosing $m=1$ is the best, it is again not GD.
>
> Finally, we highlight again, as written clearly in the introduction, that our goal and contribution lies in the theoretical analysis of adaptive gradient methods with random shuffling.

---

> > ### Comment · AnonReviewer4 · 2020-11-20
> > **Still contribution is unclear to me**
> >
> > Thank you for your reply but I still do not see any benefit of your approach.
> >
> > Lets make sure we are on the same page:
> > * T - is the total number of epochs which is proportional to the total number of computations
> > Thus you measure the performance versus total number of computations which is the right measure to use.
> >
> > *In theorem 1 you show that the "error" behaves like,
> >  $O(m^{5/4}/\sqrt{T})$ thus the optimal error is achieved when m=1 which corresponds to the case of using full gradients.
> >
> > *In case we use full gradients the random shuffling that is used is actually meaningless. Thus using full gradients is preferable compared to random shuffling together with minibatch Adaptive gradient method.
> >
> > *If we use full gradients for T epochs and the function is smooth then it is a well known result that one can obtain  O(1/\sqrt{T}) rate even without using an adaptive scheme, and simple GD suffices in this case
> >
> >
> > To conclude: you show that random shuffling with adaptive method is always worse than full-gradient with adaptive method, and in the latter case you show a results similar to simple GD. Thus, I do not see any benefit neither to random shuffling nor to using adaptive methods....
> >
> > I therefore keep my score

---

> > > ### Author Response · Authors · 2020-11-20
> > > **Appreciation for your quick response but several misunderstandings in your comment**
> > >
> > > Thank you for your quick response.
> > >
> > > For starters, we are **not** proposing a new approach. We are providing a theoretical **analysis** to gain a better understanding of **adaptive gradient methods**. Our analysis indeed gives a better convergence rate for adaptive gradient methods with random shuffling, compared with the existing rates $O(T^{-1/4})$ of adaptive gradient methods and $O(T^{-1/3})$ of random shuffling SGD.  We have stated this very clearly in the introduction as well as the first response.
> > >
> > > Second, the convergence rate of $O(m^{5/4}T^{-1/2})$ applies to any choice of $m$. $m>1$ is where random shuffling comes into meaning. One would say this result implies that choosing $m=1$ is the best, and as we discuss in the complexity analysis, this order on $m$ might not be strict and encourage future work to improve it. Because of this, we do not conclude anything on which choice of the batch size is the best. But this nevertheless does **not** affect the main contribution, which is to **improve the convergence rate of adaptive gradient methods**.
> > >
> > > Finally, it is a well-known result that GD converges with $O(T^{-1/2})$ for non-convex optimization. It is also known that vanilla SGD converges with $O(T^{-1/4})$ and random shuffling SGD convergences with $O(n^{1/3}T^{-1/3})$, which are both worse than GD.  One can hardly say these results have no contributions if they can not compete with the simplest method. It is our goal to better understand more complex methods and we believe we have achieved that.

---

### Author Response · Authors · 2020-11-20
**General Response**

We thank all the reviewers for their valuable comments, questions, and suggestions. Based on these reviews, we revise our submission into the current version.

In this part, we summarize the differences between the two versions, in an effort to answer the common hanging questions.
1. We would like to apologize for neglecting a line of works centering on the _Strong Growth Condition_ (Schmidt & Roux, 2013; Vaswani et al., 2019; Gower et al., 2019). The strong growth condition assumption is very similar to our original consistency ratio assumption in form. $$\frac{1}{n}\sum_{i=1}^n ||\nabla f_i(x)||\leq r^2 ||\nabla f(x)||^2\quad vs. \quad \frac{1}{m}\sum_{i=1}^m ||\nabla f_{B_i}(x_{t,i})||\leq r^2 ||\nabla f(x_{t,1})||^2$$
   However, the strong growth condition has much nicer properties:
   - It is algorithmic-independent in that it does not depend on $x_{t,i}$ generated by specific algorithms.
   - It can be verifiable ahead of time without simulations since it only depends on the objective function.
   - A wide range of objective functions can be shown to satisfy this condition.

2. For reasons in 1, we change our consistency ratio assumption to the strong growth condition assumption. The main analysis is almost intact with slight modifications to Lemma 1 and Lemma 3.
3. We discuss in detail functions satisfying the assumption and add in proofs to show that the assumption holds for two specific types of objective functions in section 6.
4. In the introduction, we change the epoch-wise comparison to iteration-wise comparison, which is more commonly accepted as measurements in stochastic optimization. We also make clarifications about expectation minimization and finite sum minimization for the sake of fair comparisons.
5. In experiments, since the strong growth condition is verifiable without simulations, we took out the part that tested the assumption. We point out here that the strong growth condition can be tested using experiments, but would require calculations of full gradients. Therefore, this could be a downside of the strong growth condition compared to the consistency ratio. However, since the submission focuses on theoretical results, one would prefer the strong growth condition assumption with much nicer theoretical groundings.

[1] Mark Schmidt and Nicolas Le Roux. Fast convergence of stochastic gradient descent under a strong growth condition, 2013.

[2] Sharan Vaswani, Francis Bach, and Mark Schmidt. Fast and faster convergence of sgd for over- parameterized models and an accelerated perceptron, 2019.

[3] Robert Mansel Gower, Nicolas Loizou, Xun Qian, Alibek Sailanbayev, Egor Shulgin, and Peter Richtarik. SGD: General analysis and improved rates, 2019.

---

### Decision · Program_Chairs · 2021-01-07
**Final Decision**

**Decision:**

Reject

**Comment:**

Dear authors,

Improving the theoretical understanding of powerful algorithms is an important contribution to our field. Nevertheless, most of the reviewers are inclined to reject the paper. I somehow have to agree with them as e.g., adding more restrictive assumptions can allow deriving better bounds, but the question then is how useful this result will be to the ICLR community. I would encourage you to chose maybe another venue.

Thanks